# BRain health and healthy AgeINg in retired rugby union players, the BRAIN Study: study protocol for an observational study in the UK

Valentina Gallo,[1,2,3] Damien McElvenny,[4] Catherine Hobbs,[2] Donna Davoren,[2] Huw Morris,[5] Sebastian Crutch,[6] Henrik Zetterberg,[7,8,9,10] Nick C Fox,[6] Simon Kemp,[11] Matthew Cross,[11] Nigel K Arden,[12] Madeleine A M Davies,[12] Andrea Malaspina,[13] Neil Pearce[2]

For numbered affiliations see end of article.

**Correspondence to**
Dr Valentina Gallo;
v.gallo@qmul.ac.uk

## ABSTRACT

**Introduction** Relatively little is known about the long-term health of former elite rugby players, or former sportspeople more generally. As well as the potential benefits of being former elite sportspersons, there may be potential health risks from exposures occurring during an individual's playing career, as well as following retirement. Each contact sport has vastly different playing dynamics, therefore exposing its players to different types of potential traumas. Current evidence suggests that these are not necessarily comparable in terms of pathophysiology, and their potential long-term adverse effects might also differ. There is currently limited but increasing evidence that poorer age-related and neurological health exists among former professional sportsmen exposed to repetitive concussions; however the evidence is limited on rugby union players, specifically.

**Methods and analysis** We present the protocol for a cross-sectional study to assess the association between self-reported history of concussion during a playing career, and subsequent measures of healthy ageing and neurological and cognitive impairment. We are recruiting a sample of approximately 200 retired rugby players (former Oxford and Cambridge University rugby players and members of the England Rugby International Club) aged 50 years or more, and collecting a number of general and neurological health-related outcome measures though validated assessments. Biomarkers of neurodegeneration (neurofilaments and tau) will be also be measured. Although the study is focusing on rugby union players specifically, the general study design and the methods for assessing neurological health are likely to be relevant to other studies of former elite sportspersons.

**Ethics and dissemination** The study has been approved by the Ethical Committee of London School of Hygiene and Tropical Medicine (reference: 11634-2). It is intended that results of this study will be published in peer-reviewed medical journals, communicated to participants, the general public and all relevant stakeholders.

## INTRODUCTION

The evidence relating to head trauma in sport and the subsequent risks of neurological disease have been previously been reviewed,

## Strengths and limitations of this study

► The study will provide meaningful data on the burden of health and neurological health among retired rugby players (in only 2 years).
► The cross-sectional design does not prevent the potential for recall bias and selection bias.
► Results may not be immediately generalisable to current players as in the last 30 years playing rules and conditions have changed.

and it has been established that sports involving repeated head trauma may have an increased risk of neurodegenerative disease in the long term.[1] Furthermore, there are now plausible mechanisms for these effects, and a recognition that these problems do not just occur in former boxers, but in a variety of sports involving repeated concussions,[2] and possibly also in sports in which low-level head trauma is common.[3] These neurodegenerative effects include potentially increased risks of impaired cognitive function and dementia,[4–7] Parkinson's disease (PD)[8–13] and amyotrophic lateral sclerosis (ALS).[3 14–19] The term *chronic traumatic encephalopathy* (CTE) was introduced as a clinicopathological construct for the neurodegeneration associated with American football and wrestling[20] (see table 1).

Each contact sport has vastly different playing dynamics thereby exposing its players to different types of potential traumas. Current evidence suggests that these are not necessarily comparable in terms of pathophysiology, and hence in terms of their potential long-term adverse effects on health. There is currently limited but increasing evidence that poorer general and neurological health exists among professional sportsmen exposed

**Table 1** Glossary of definitions, adapted from Jordan[32]

| | |
|---|---|
| Chronic traumatic brain injury (TBI) | A spectrum of disorders associated with long-term consequences of single or repetitive TBI. |
| Chronic traumatic encephalopathy (CTE)* | Prototypic chronic TBI, long-term neurological consequences of repetitive mild TBI. |
| Dementia pugilistica | A subtype of CTE that is typically reserved for cases of severe end-stage dementia secondary to a long boxing career. |
| Chronic postconcussion syndrome* | A condition in those athletes in whom post-concussion symptoms do not appear to resolve. |
| Chronic neurocognitive impairment* | A rather diverse classification of chronic neurocognitive signs and symptoms secondary to head impact exposures and recurrent concussions that is theoretically distinctive from CTE. |
| Post-traumatic dementia† | Cases in which the athlete meets clinical criteria for dementia secondary to a single moderate-to-severe TBI. |
| Post-traumatic cognitive impairment† | Individuals who sustain long-term neurocognitive deficits from a single moderate-to-severe brain injury and do not meet clinical criteria for dementia, but instead mild cognitive impairment. |
| Post-traumatic parkinsonism | A Parkinsonian-like syndrome secondary to a single moderate-to-severe or repetitive TBI, occurring solely or as a component of CTE. |

*The most clinically pertinent examples of sports-related chronic TBI.
†Consequence of a single brain injury.

to repetitive concussions; however, there is little evidence from rugby union players.[21–23]

Decq et al[21] investigated retired French-speaking high-level sportsmen, aged 45–65 years, who had played sports for at least 10 years. Mild cognitive disorder was lower in players of other sports (40.4%) than in former rugby players (56.6%) (P=0.005). However, after adjustment for smoking and higher education, no association was observed between cognitive function and number of reported concussions.[21] In New Zealand, 366 former players were tested on their engagement in sport, general health, sports injuries and concussion history, demographic information and cognitive functioning. The elite rugby group performed worse on tests of complex attention, processing speed, executive functioning and cognitive flexibility than the non-contact sport group, and worse than the community rugby group on complex attention. Former players who recalled one or more concussions had worse scores on cognitive flexibility, executive functioning and complex attention than players who did not recall experiencing a concussion.[22] A recent Scottish study assessed 52 former Scottish international rugby players and 29 controls. Players performed worse on a test of verbal learning, and of fine coordination of the dominant hand; however, no statistically significant differences were observed on other cognitive tests. Additionally, no significant association was found between the number of concussions and cognitive test performance.[23]

Ultimately, establishing the extent of these potential issues will require long-term prospective studies involving the repeated measurement of head trauma exposures and repeated tests of neurological health in large numbers of current and former players. However, a first step in this process is to conduct cross-sectional studies in former players to assess whether there is an association between their history of concussion during their rugby careers, and subsequent measures of healthy ageing and subtle neurological and cognitive impairment. In particular, such cross-sectional studies can be conducted in a relatively short time and with 'standard' measures of cognitive function since impaired cognitive function is an important health outcome in itself, and may be a precursor of more serious long-term neurological effects.[24]

### Definitions of trauma and concussion

Traumatic brain injury (TBI) is usually classified as mild, moderate or severe, on the basis of the initial Glasgow Coma Scale[25] recorded in emergency departments including the duration of any loss of consciousness; and duration of post-traumatic amnesia (ie, loss of memory of events after the injury).[26] Chronic TBI represents a spectrum of disorders associated with long-term consequences after single or repetitive TBI.[27] Conversely, there remains limited consensus on the definition of concussion. The 2012 Zurich Consensus Statement on Concussion in Sport proposed that concussion and mild TBI should be viewed as distinct entities.[28] The group defined concussion as a 'complex pathophysiological process affecting the brain', and despite allowing for the presence of neuropathological damage, they postulated that concussive symptoms largely reflected a functional disturbance, typically resolving spontaneously with no imaging abnormality. In contrast, recent guidelines from the American Academy of Neurology for sports concussion in 2013, do not separate concussion from mild TBI, defining concussion as 'a clinical syndrome of biomechanically induced alteration of brain function, typically affecting memory and orientation, which may involve loss of consciousness'.[29] A recent report[30] examines why having two different pathological entities might be unhelpful and suggests that the Mayo

Clinic TBI Classification system should cover both definitions.[31] A glossary of definitions used in this protocol is found in table 1, adapted from Jordan.[32]

For the purpose of this study, we adapted the National Health Institute (NIH) definition of concussion.[33] Participants will be asked to report their previous concussions according with the following definition:

Concussion is defined as an alteration in brain function, caused by an external force. Symptoms include:
► A decreased level/loss of consciousness
► Memory loss (before or after the injury)
► Weakness
► Temporary paralysis
► Loss of balance
► Change in vision (eg, blurriness, double vision)
► Coordination difficulties
► Numbness
► Decreased sense of smell
► Difficulty understanding what others are saying
► Difficulty communicating with others
► Confusion, disorientation or slowed thinking

Please note, loss of consciousness is not required for a concussion to be diagnosed.

## Biomarkers

Research in the field of TBI biomarkers has increased exponentially over the last 20 years,[27 34–36] with studies assessing biomarkers that could provide diagnostic and prognostic, as well as monitoring information.[37 38] However, to date, no biomarker for the long-term effects of concussion has been identified, although recent studies on cerebrospinal fluid (CSF) from sportsmen with post-concussive symptoms suggest that a subset of these have biomarker signs of ongoing axonal injury and microglial activation.[39–41]

Of the biomarkers measurable in serum or other body fluids, those more likely to be detectable for longer time periods after the concussion episode, are those related to the axonal injury and more in generally neuroinflammation, that is neurofilament (NF) and tau protein.[38] NFs coassemble from protein subunits to form NFs defined as light (light (NF-L), medium (NF-M) or heavy (NF-H) according to their relative molecular weights). NFs are one of the key structural elements of neurons, providing mechanical stability and determining axonal diameter. NFs are of particular interest because as they are structural elements, they may be more susceptible to mechanical deformation under the condition of trauma. Abnormal NF aggregation may contribute to the delayed or progressive neuronal death and dysfunction taking place in neurodegenerative diseases associated with NF aggregation, such as PD, ALS and Lewy body dementia.[42–44] Acute perturbations of NFs have been demonstrated in experimental models of TBI that produce cortical contusion in combination with selective hippocampal neuronal death.[45] A study compared serum NF-L, a biological marker of head trauma, in American football athletes and non-contact sport athletes and examined changes over the course of a season. Results suggest that a season of collegiate American football is associated with elevations in serum NF-L, which is indicative of axonal injury, resultant of head impacts.[46] NF-L has also been associated with head trauma severity detected by CT scan, immediately after the episode[47] and with concussive and subconcussive head impacts in boxing.[41] Importantly, while plasma tau concentration increases and disappears rapidly (within hours to a few days) following concussion,[48] NF-L has prolonged increased levels lasting for many weeks.[41 49 50]

Phosphorylation of tau is a normal event in healthy neurons, but hyperphosphorylation and aggregation into neurofibrillary tangles is a characteristic of Alzheimer's disease and CTE.[51] Tau concentrations correlate with lesion size and outcome in severe TBI when measured in the ventricular CSF, while remaining unchanged in TBI and other forms of acute brain injury.[38] Studies on mild TBI show increased CSF concentrations of both t–tau and NF–L, although the increase in CSF NF-L is greater than the increase in t-tau, suggesting that head impacts have a greater effect on long, large-calibre axons that extend subcortically than on short, non-myelinated axons in the cortex.[38] After concussion, serum tau concentrations were found to be increased, but this increase did not correlate with severity of trauma and lesion load as measured using CT scan signs.[38]

## Genetics

Few hypotheses on the potential role of genetics in modulating the possible association between concussion and cognitive decline have been formulated. Specific genes might be associated with an increased risk of concussion via worse attention or executive function, or more vulnerable brain anatomy,[52] or via personality trait.[53] Conversely, a number of genes are involved in modulating the risk of developing dementia and other neurodegenerative diseases, irrespective of concussion. Dementia risk is higher among carriers of the epsilon4 allele at the apolipoprotein E gene.[54] A number of genome-wide association studies (GWAS) have been performed to date, enabling the identification of 24 loci as risk factors for PD.

## Rationale of the BRain health and healthy AgeINg (BRAIN) Study

Recently, World Rugby (formally known as the International Rugby Board), has developed a process to support team clinicians in the recognition, assessment and subsequent management of elite adult players who have sustained a potential concussion. This process includes the development of a multimodal assessment: the Head Injury Assessment, formerly the Pitch Side Concussion Assessment tool.[55]

However, there is little clear evidence, about the possible long-term effects of concussion in rugby union players, and it is not clear if findings from other sports are directly generalisable to rugby players. Many sports involve exposure to concussion or repetitive low-level head trauma, and it can be argued that each sport should

be considered independently, due to the unique technical and physiological profile that a player develops over the course of a career.[56 57] Thus, to determine the potential risk of long-term adverse health effects of playing rugby specific studies of rugby players are needed.

The BRAIN Study builds on a recent cross-sectional questionnaire-based study conducted by the University of Oxford as part of the Arthritis Research UK Centre for Sport, Exercise and Osteoarthritis, who have assessed the general and musculoskeletal health of former elite rugby players.[58] Rugby players from the Oxford and Cambridge University Football Clubs (Oxbridge Blues), and members of the England Rugby Internationals Club—a membership organisation of all current and former England players—were recruited. Self-reported demographic factors, playing history (including head trauma), past medical history (including dementia, depression and memory impairment), and perceived health were collected, in addition to detailed information on musculoskeletal health and pain.

A total of 319 participants have participated in this study to date, and 205 of those aged 50 years or older have agreed to be contacted again for further studies. The 205 participants aged 50+ years who agreed to be contacted again will be invited to take part in the BRAIN Study.

## Aim and objectives

The overall aim of the BRAIN Study is to investigate the possible associations between concussion in rugby and ageing, including physical and cognitive capabilities, as well intermediate neurological and musculoskeletal end points among former rugby players.

In order to achieve these aims the following objectives will be pursued:

1. To investigate the associations between self-reported concussion history and ageing, measured as physical and cognitive capabilities. This will be achieved by using the following outcome measures:
   a. Physical capability outcomes: grip strength, chair raise and walking speed.
   b. Cognitive capability outcomes: memory, reasoning, speed of thinking and attention, and verbal and numerical skills.
2. To investigate the association between self-reported concussion history and intermediate neurological end points
   a. Neurological examination: a brief neurological examination will be video-recorded according to existing protocols and independently examined by two neurologists.
   b. Intermediate neurological end points: information using tapping test (BRAIN), smell test, REM-Behaviour Disorder (RBD; REM, rapid eye movement) Questionnaire will be collected in order to explore non-motor symptoms of PD and related disorders.
3. To investigate whether a history of concussion is associated with current tau protein and NF-L levels in blood (potential biomarkers of neurodegeneration), and how these biomarkers are in turn associated with the outcome measures (potential biomarkers of early detection of disease).
4. To assess if any other characteristic of rugby playing history, in addition to self-reported concussion (ie, length of playing at elite level, position of play, number of games played, age when started playing), or age at concussion is associated with any outcome measure (physical and cognitive capability and/or intermediate neurological outcomes).
5. To investigate the long-term musculoskeletal health outcomes of rugby players with particular emphasis on hip, knee and hand osteoarthritis allowing changes over time (from the previous study to the current study).

In addition, we will develop a multimedia database (data from tests and questionnaire plus video-recorded neurological examination plus biobank of blood samples) that will serve as baseline for further tests and further follow-up.

## Study design

Participants will be invited to study clinics in London, Manchester or Bath. A home visit can also be arranged by request. Participants will be invited to bring with them any medication they are taking regularly for an accurate recording of current medication usage.

All participants will be administered a *Core Module* interview including questions on lifestyle factors, potential confounders, and extensive information on the five domains of physical and cognitive ability, neurological examination, intermediate neurological outcomes and musculoskeletal health.

The lifestyle questionnaire will complement that already administered to participants of the Oxford-based cross-sectional study, and will include questions on potential confounders of the association between concussion and physical and cognitive capability, and intermediate neurological outcomes. Participants will also be asked to donate a blood sample (a normal blood sample will be collected by participants seen in one of the clinics, a dry spot sample plus saliva swab for DNA will be collected from participants seen at home). At the end of the Core Module, all participants will also be asked to undergo some additional tests (the *Additional Optional Module*) provided they have sufficient time (figure 1). At the end of the interview, all participants will be asked some final questions on their concussion history. This will ensure that the interviewer is blind to the participant's concussion history during the entire duration of the interview, and will also act as validation to confirm data previously collected. Participants who disclose information about their concussion history during the interview, will be noted in order to undertake a sensitivity analysis excluding them.

The Core Module includes: (1) a lifestyle questionnaire; (2) a set of tests covering essential information on physical capability (height, weight, grip strength, chair

| | Questionnaire | Physical ability | Cognitive ability | Neurological examination | Intermediate neurological outcomes | Blood sample |
|---|---|---|---|---|---|---|
| 'Core' Module | •Lifestyle and confounders questionnaire<br>•Concussion history questionnaire≠ | •Height<br>•Weight<br>•Grip strength<br>•Chair rise^<br>•Walking speed#<br>•Photograph of hands | •MMSE<br>•WMS-R logical memory<br>•FNAME-12A<br>•Task-set shifting/Response inhibition<br>•WAIS-R digit symbol<br>•Visual short-term memory binding<br>•NART | •Video-recording~ | •BRAIN test | •DNA*<br>•Plasma* |
| Additional Optional Module | •Pain mannequin (hand) | • HOOS<br>•KOOS<br>•QuIKS | •Visuomotor integration<br>•Matrix reasoning (WASI)<br>•Irrelevant Distractor Paradigm | •UPDRS-II | •Smell test<br>•RBD questionnaire | |

**Figure 1** Framework for data collection in the BRAIN Study by domain. *not to be collected if the participant is seen at home; ^subject to suitable chair availability for the participants seen at home; ~including knee bending test; #subject to space availability if the participant is seen at home; ≠to be administered as last item in all cases (after all 'Core' and Additional Optional Module tests). BRAIN, BRain health and healthy AgeIng; FNAME-12, 12-item Face-Name Associative Memory Exam; HOOS, Hip Disability and Osteoarthritis Outcome Score; KOOS, Knee Injury and Osteoarthritis Outcome Score; MMSE, Mini-Mental State Examination; NART, National Adult Reading Test; QuIKS, Questionnaire to Identify Knee Symptoms; RBD, REM-Behaviour Disorder; WAIS-R, Wechsler Adult Intelligence Scale-Revised; WASI, Wechsler Abbreviated Scale of Intelligence; WMS-R, Wechsler Memory Scale-Revised.

rise, walking speed and photo of the hands) and cognitive capability (Mini-Mental State Examination (MMSE), Logical Memory, Digit-Symbol Substitution test, Matrix Reasoning, Task-set Shifting/Response Inhibition, Visuomotor Integration, 12-item Face-Name Associative Memory Exam (FNAME-12) and the National Adult Reading Test (NART)); (3) a remote neurological clinical examination (video-recorded); and (4) a test for subtle movement disorders, the BRAIN tapping test. This Core Module should take no longer of 1 hour and 45 min to be completed.

The Additional Optional Module includes: (1) a questionnaire investigating hand pain; (2) extra tests investigating the cognitive domain (Visual short-term memory binding, Irrelevant Distractor Paradigm); 2) the Unified Parkinson's Disease Rating Scale (UPDRS) Part II Scale to complete the neurological examination (not video-recorded); (3) additional tests for intermediate neurological outcomes (the smell test and the RBD Questionnaire). This Additional Optional Module should take no longer than an hour and half to be completed (figure 1). The full test/interview including core and additional modules should not take more than 3 hours and 15 min to complete.

## General questionnaires

Questionnaires will be used to collect relevant information on:

► Lifestyle and confounders - All participants will be asked a number of questions on their lifestyle in order to collect information on possible confounders for the main analysis (smoking, alcohol, coffee, drugs, past medical history, sleep quality).

► Concussion history - At the end of the interview, participants will again be asked some detailed information regarding their history of head trauma and concussion while playing.

► Musculoskeletal hand pain - A questionnaire on hand pain to identify possible hand osteoarthritis will be used involving a hand manikin (Additional Optional Module).

## Physical ability assessment

The methodology used in the 1946 birth cohort[59] will be leveraged to assess physical and cognitive capabilities in this study. This will enhance comparability of results and facilitate future collaborations. The most commonly used objective measures of physical capability for assessing healthy ageing, are tests of grip strength,[60 61] walking

speed,[60 62 63] chair rises[64] and standing balance;[60] these aim to assess physical functioning, including the capacity to undertake the physical tasks of daily living.[65] There is robust evidence that higher scores on these measures are associated with lower rates of mortality, and there is more limited evidence of lower risks of morbidity, and of age-related patterns of change.[65]

Height and weight– Height and weight will be collected in order to calculate the body mass index (BMI) according to the formula BMI=weight (kg)/height (m)$^2$.

Grip strength- JAMAR hydraulic hand dynamometer is the most widely used instrument with established test-retest, inter-rater and intrarater reliability.[66]

Chair rise- The 30s Chair Stand Test will be used to assess lower body strength.[67] This test provides a reliable and valid indicator of strength in adequately active, older adults form the general population. Intraclass correlation coefficients for test-retest are 0.84 for men and 0.92 for women, using one-way analysis of variance, and a non-significant change in scores between days indicating good reliability.[67] As expected, chair-stand performance decreased significantly with increasing age (P<0.01) and was statistically significantly lower for participants who were less active (P<0.0001).[67]

Walking speed– In order to assess normal comfortable walking speed and maximum walking speed, a timed 10-Metre Walk Test will be used. This test has been validated in 230 healthy volunteers.[62] The mean comfortable gait speed ranged from 127.2 cm/s (women aged 70 to 79 years) to 146.2 cm/s (men aged 40–49 years). Mean maximum gait speed ranged from 174.9 cm/s (women aged 70–79 years) to 253.3 cm/s (men aged 20–29 years). The correlation coefficients of both gait speed measures was higher than 0.903, and correlated significantly with age (r>−0.210) and height (r>0.220).[62]

A photograph of the hands will be taken at the end of this section to assess inflammation, finger nodes, and other rheumatological and osteoarthritis-related changes at the hand.

The Hip Disability and Osteoarthritis Outcome Score measures patient's opinions about their hip and associated problems. It examines pain, symptoms, function in activities of daily living (ADL) and function in sport and recreation. It has been used in subjects with hip disability with or without hip osteoarthritis[68] (Additional Optional Module).

The Knee Injury and Osteoarthritis Outcome Score examines pain frequency and severity during functional activities, and symptoms such as the severity of knee stiffness, the presence of swelling, grinding or clicking, difficulty in ADL, with sport and recreation, and knee-related quality of life. It is intended for use in young and middle-aged populations with post-traumatic osteoarthritis, in addition to those with injuries who may go on to develop secondary osteoarthritis[69] (Additional Optional Module).

The Questionnaire to Identify Knee Symptoms is used for early osteoarthritis, and understanding symptomology

and potential adaptation to activity before osteoarthritis has been clinically diagnosed. It has been developed in adults aged 40–65 years with evidence of ongoing knee problems and recommended for use in studies exploring early osteoarthritis[70] (Additional Optional Module).

## Cognitive ability assessment

The methodology used in the 1946 birth cohort will also be followed for the assessment of cognitive capability.[59] Tests and questionnaires will be described according their belonging to the Core Module or the Additional Optional Module, and by domain (figure 1).

*The MMSE*[71] is a widely used 30-point screening tool for cognitive impairment within clinical practice, assessing multiple cognitive domains including orientation to time and place; registration; attention ±calculation; recall; language; repetition; reading; writing; visuospatial function; and executive function and praxis.

The Logical Memory from the Wechsler Memory Scale-Revised[72] test assesses free recall of a short story. The participant is asked to recall the story immediately and after 20 min delay.

FNAME-12A is a modified version of the 16-item Face-Name Associative Memory Exam (FNAME-16). FNAME-12A has fewer stimuli and additional learning trials which have been found to be well tolerated by those with mild cognitive impairment, while remaining challenging in cognitively normal older adults.[73] FNAME-12A has demonstrated psychometric equivalence with FNAME-16, which has been shown to be related to the beta-amyloid burden in cognitively normal elderly people.[74] FNAME-12A requires the participant to learn 12 face-name and face-occupation pairs. Participants are given two exposures to all 12 face-name/occupation pairs. After each exposure, and following 5 min delay, participants are asked for the name and occupation associated with each face. After 30 min delay they are shown three faces and asked to identify the face that they recognise and give the name and occupation. Given multiple options to choose from, they are then asked to select the name and/or occupation associated with the face.

The Task-set Shifting/Response Inhibition[75 76] task examines the relationship between executive tasks of task-switching/preparation time. In the arrow only condition, participants are shown the cue 'arrow'. Following a short delay they must respond to the direction of the arrow ('right' or 'left'). In the word only condition, participants are shown the cue 'word'. Following a short delay they must respond to the direction of the word ('right' or 'left'). There is a switching condition in which the participant is shown the cue 'arrow' or 'word'. Following a delay, both a combined arrow and word stimulus appears. The stimulus is either congruent (left arrow and left word), or incongruent (left arrow and right word). Trials in the switching task are categorised into switch and non-switch. In a non-switch trial the cue is the same as for the immediately preceding trial. In a switch trial the cue differs from the immediately preceding trial.

The Digit-Symbol Substitution Test, from the *Wechsler Adult Intelligence Scale-Revised*[77] explores attention and psychomotor speed. Participants are given a code table displaying digits (from 1 to 9); each digit is paired with a symbol. The participant is required to fill in blank squares with the corresponding symbol for each digit as shown in the code table. They are given 90 s to fill in as many squares as possible.

The Visual Short-term Memory Binding[78] [79] test requires the participant to view one or three fractal objects, presented simultaneously in random locations on the screen. The participant is asked to remember both the objects and their location. After a delay of 1 s or 4 s they have to make a forced choice between one of the displayed fractals (the target) and a 'dummy' fractal. Participants are required to touch the object they think has been previously presented and 'drag' it on the touch screen to its remembered, original location. The binding of such featured information has been shown to be vulnerable in asymptomatic FAD mutation carriers.[79]

NART was specially designed to provide a means of estimating the premorbid intelligence levels of adults suspected of suffering from intellectual deterioration.[80] NART comprises a list of 50 words printed in order of increasing difficulty. The words are relatively short in order to avoid the possible adverse effects of stimulus complexity on the reading of dementing subjects, and they are all 'irregular' with respect to the common rules of pronunciation in order to minimise the possibility of reading by phonemic decoding rather than word recognition.[80]

Visuomotor Integration[81] is a circle tracing task which includes both direct and indirect visual feedback conditions. Continuous performance measures are provided including accuracy, speed, and speed of error detection and correction. The test has revealed changes in speed and accuracy in Huntington disease mutation carriers more than 10 years before expected age-of-onset (Additional Optional Module).

The Matrix Reasoning from the *Wechsler Abbreviated Scale of Intelligence*[82] test assesses non-verbal reasoning. The participant is shown a matrix of geometrical shapes with a section missing. They are required to select the option that completes the matrix (Additional Optional Module).

In the irrelevant distractor paradigm[83] participants are given a computerised letter-search task and are required to make a rapid decision as to whether the target letter 'X' or 'N' has appeared in the search display (in either low or high load conditions). On some of the trials, a task-irrelevant distractor (a cartoon character) appears on the outside of the search display. The task evaluates the extent to which attention is captured and captivated by the distractor (Additional Optional Module).

## Neurological clinical assessment

The neurological assessment of participants is needed for detecting any subtle neurological sign, and to establish a baseline for the absence of one or more signs. A video-recorded standard neurological examination will be included as part of the Core Module. An additional test of self-reported impairment, mainly due to movement disorders, will be administered as part of the Additional Optional Module (figure 1).

A standard video recording of each participant accessing the study at one of the clinics will be performed and evaluated by a clinical neurologist. The assessment includes examination of strength, coordination, balance, ocular movements, cranial nerves, gait and repeated movements. The examination also includes a test of knee functionality.[84]

Information on potential movement disorders will be collected using the Unified Parkinson's Disease Rating Scale, part II. This includes self-reported ratings on several motor domains, including speech, saliva and drooling, chewing and swallowing, eating, dressing, hygiene, hand-writing, hobbies, turning in bed, tremor, standing up, walking and balance, and freezing (Additional Optional Module).

## Intermediate neurological end points

In addition to neurological signs and symptoms, some intermediate outcomes have been recognised as part of the complex clinical picture of movement disorders, manifesting before the onset of the movement impairment itself. These will be investigated with three tests described below (figure 1).

The BRAIN tap test is a simple computerised test which uses a standard keyboard, based on alternating finger tapping. It has been developed to quantify and define upper limb motor function.[85] The test generates four variables: (1) the Kinesia Score which is defined as the number of keystrokes in 60 s; (2) the akinesia time which is defined as the cumulative time that keys are depressed; (3) the Dysmetria Score which is calculated as a weighted index using the number of incorrectly hit keys corrected for speed; and (4) the In-coordination Score, which is a measure of rhythmicity corresponding to the variance of the time interval between keystrokes.[85] The BRAIN test assesses speed, accuracy and rhythmicity of upper limb movements, regardless of their causes. The results of the test correlate well with PD clinical rating scales and with cerebellar dysfunction.[85] The BRAIN tap test is available online at https://www.braintaptest.com/en_GB and VG has obtained access for generating tokens for administering the test to participants (A Noyce, personal communication).

The University of Pennsylvania Smell Test is a measurement of the individual's ability to detect odours.[86] The test, which takes only a few minutes, consists of four different 10-page booklets (total of 40 questions). On each page, the participant is invited to sniff a different 'scratch and sniff' strip, embedded with an odorant. For each odour, the participant can choose among four multiple-choice answers. The total score of each participant is compared with scores in a normative database from 4000 normal individuals[87] (Additional Optional Module).

The RBD Screening Questionnaire (RBDSQ) is a validated 10-item patient self-rating questionnaire (maximum total score 13 points) covering the clinical features of RBDs.[88] The RBDSQ was validated on 54 patients with polysomnographically confirmed RBD, 160 control subjects in whom RBD was excluded by history and polysomnography and 133 unselected healthy subjects. Applying a positivity threshold of 5 points, RBDSQ had a sensitivity of 0.96 and a specificity of 0.56 comparing patients with RBD with patients with other sleep disturbances. A specificity of 0.92 was calculated when comparing patients with RBD with patients without sleep disturbances[88] (Additional Optional Module).

### Blood and saliva sample collection, and biobanking

Participants seen in the research clinics will be asked to donate a blood sample for testing for tau protein, NF levels, full blood count analysis and to obtain data for GWAS. Samples of serum, plasma and DNA will be collected and processed according to a prespecified protocol and stored in a freezer at −80°C at the Blizard Institute (Queen Mary, University of London). All participants will also asked to donate a blood sample through a dried blood spot test and a saliva sample (in order to obtain DNA where phlebotomy is not possible).

Samples will be tested to measure the expression of NF-L, an NF subunit showing consistent and reproducible measurements in plasma/serum using a newly developed sensitive methodology exploiting single molecule trapping and measurement by ELISA Single Molecule Array (SIMOA).[89–91] The use of both NF-L and NF-H allows for the determination of different markers whose dynamics of release into biofluids following injury vary according to their particular chemical structure. NF-L has shown good levels of linearity on dilution experiments, suggesting that confounders like aggregation or immune response may not interfere with its measurements. The different NF-H phosphorylation states in blood may in turn provide more information on the systemic biological changes induced by trauma. The relative low abundance of these proteins in blood, in the absence of CSF, which is normally more enriched with the by-products of neuronal destruction, will not represent a major obstacle, as analytical sensitivity has evolved with the development of novel assays which cover the lower end of these dynamic protein ranges. SIMOA, the proposed methodology for the target biomarkers, is now emerging as the core technique for the measurement of structural proteins, down to the femtomolar end of the spectrum. A novel SIMOA-based assay for the measurement of tau in blood is now available. This development offers a wide range of opportunities based on the possibility of overcoming the limitations imposed by CSF collection by lumbar puncture.

GWAS will be conducted at University College London (UCL) in the lab led by Prof John Hardy.

### Multimedia database set-up and management

A multimedia database will be established as part of this project. The final data set will include questionnaire and tests data (see figure 1), in addition to the video-recorded neurological examination. These data will be also linked to the stored blood samples through a unique identifier.

The data will be stored in mirror, anonymised data sets at the London School of Hygiene and Tropical Medicine and Queen Mary, University of London (QMUL) on secure servers, and access will be regulated by the study Principal Investigators (PIs) who are ultimately responsible for maintaining confidentiality and data protection. Only the researcher(s) employed on this study will have access to subject identifiable information. Identifiable information (name, date of birth and address) will be removed prior to analysis and subjects will be identified by pseudoanonymised codes. Personal data and key codes for pseudoanonymisation will be stored in a database with restricted access. Before conducting the statistical analyses, checks will undertaken to ensure data quality.

### Statistical analysis

Descriptive statistical methods (geometrical means, geometrical SDs and 95% CIs) will be used to summarise the rugby history and lifestyle exposure data. Analyses linking a history of head trauma with neurological symptoms (obtained by questionnaire and neurobehavioural tests) will then be performed. Parameters of physical and cognitive capability, alongside clinical neurological intermediate end points, will initially be dichotomised (yes/no symptoms), and analysed with prevalence ORs using logistic regression.[92] We will also analyse continuous outcome measures using multiple linear regression. In addition to analysing individual symptoms, we will assess associations with domains of symptoms (ie, physical and cognitive capability). For this purpose we will use cut points previously published in the literature,[93] particularly related to the 1946 birth cohort whose testing protocol was used. For analyses involving repeated measurements (neurobehavioural testing) assessing acute and chronic effects, we will use multilevel models. Analyses will be adjusted for age, sex and alcohol consumption.

Finally, we will compare neurological outcomes assessed by questionnaire with those assessed by computer-administered tests using kappa-statistics (and in case of continuous outcomes linear regression analyses). We will also compare the strengths of the associations with exposure between both outcome measures.

### Study size and power

We aim to invite 205 participants to participate, approximately half of whom have been exposed to concussion during their career (M Cross, personal communication). We conservatively estimate that about 150 former players will participate. Based on previous studies, the SDs of the psychometric tests are in the range of 8%–15% of the absolute value; assuming a conservative figure of

15% overall, the study will have more than 95% power to detect a 10% difference, and 80% power to detect a 7% difference in psychometric test scores between exposed (to concussion) and non-exposed participants.

## Limitations

The main limitation of this study is the cross-sectional design. Data on lifetime exposure (history of concussion) and health outcomes are collected at one time point, making data subject to recall bias. Participants with a poorer health, or sentiment of discontent with their health status, may overestimate concussive exposure, if they attribute their poor health to it.

However, this cross-sectional study design will provide meaningful data on the burden of ill health and neurological health among retired rugby players, in a relatively short 2-year time frame. Depending on results (and availability of additional funding), this study could inform a more detailed, and time-intensive cohort study, aimed at assessing the association between concussion in rugby players and health outcomes prospectively.

A direct consequence of the cross-sectional design is an intrinsic risk of selection bias. Rugby players who have developed serious neurodegenerative conditions (or have died from them) in the meantime are less likely (or not able) to participate in the study thus limiting our sample to a subsample of healthier retired rugby players. On the other hand, it is possible that former players who experience some cognitive symptoms will be more keen to participate in the study, to seek reassurance and be examined in more detail.

Irrespective of these considerations, our investigation of the association between the exposure to sporting concussion and later-life general and neurological health outcomes remains valid, despite a smaller range of health outcomes diminishing the power for detecting an association as significant. Nonetheless, the power calculation for this study is calculated on the basis of continuous outcome measures (ie, neurocognitive tests or grip strength measurement) making this study adequately powered to detect the potential associations, based on the expected number of recruited participants.

It is worth noting that results from this study might be not immediately applicable to current rugby players, due to this cohort's former playing status, history of amateur playing exposure and alterations to rules and conditions, aimed at increasing player safety over recent years, which will have altered the player's overall exposure to concussive events.

## Ethics and dissemination

The study in general is focusing on subclinical problems (eg, slightly lower than average scores in cognitive function tests) which would not usually require any medical attention. If any problems are identified, we will (with the permission of the study participant) refer participants to their primary care physician. The consent form includes the following statement:

I wish to inform my GP that I am taking part to this study

I wish the study team to inform my GP of any unusual test result

This study will provide unique insight into the physical and neurological health of retired rugby players who are now aged 50 years or more. This is valuable information per se, potentially comparable with other occupational cohorts or the general population (accepting the potential for bias discussed previously in comparing these samples). Moreover, within this study, data will be analysed as a function of the history of TBI and concussion during their careers, and therefore assess associations between the history of reported concussive injuries and health-related outcomes.

Importantly, the neurological examination and collection of intermediate neurological outcomes allows the detection of subtle preclinical changes that might be associated with future risk of developing a neurodegenerative condition. Studying to what extent these changes are also associated with history of concussion will allow an estimation of the impact of concussion on the onset of neurodegenerative diseases in rugby players, thus providing the basis for future prospective studies.

A novel aspect of this project is the molecular component: clinical and neurocognitive findings will be particularly reinforced by the measurement of blood biomarkers. These will give a quantitative measurement of active neurodegeneration (NF-L) and of possible CTE (tau), to be analysed conjunctly with the health-related outcomes and personal history of concussion.

Also, relying on a number characteristics of the rugby playing history collected (ie, length of playing at elite level, position played, numbers of games played, age when started playing), it will be possible to identify potential categories of players at potentially increased risks of health-related outcomes.

This study may identify a small deviation from normality in any of the continuous measures collected via cognitive and other tests (ie, grip strength, memory function, walking speed, etc). This will be mainly investigated in association with a history of frequent concussions, or other head traumas. The investigators will interpret the results based on the continuum of distribution of the test outcomes, which will not have specific clinical translation. However, it should be emphasised that although this study is unlikely to identify patterns suggesting a clinical disease in any of the participants, if these are identified, then the relevant participant(s) will be referred to appropriate medical services.

These data will be disseminated initially to players and sports governing bodies, and through peer-reviewed publication and conference presentation, in order to begin to establish an evidence basis for the association between exposure to sports-related concussion, and potential later-life cognitive and neurological impairment.

**Author affiliations**
[1]School of Public Health, Imperial College London, London, UK
[2]Epidemiology and Medical Statistics, London School of Hygiene and Tropical Medicine, London, UK
[3]Centre for Primary Care and Public Health, Queen Mary, University of London, London, UK
[4]Research Division, Institute of Occupational Medicine, Edinburgh, UK
[5]Department of Clinical Neuroscience, University College London, London, UK
[6]Dementia Research Centre, UCL Institute of Neurology, University College London, London, UK
[7]Department of molecular neuroscience, UCL Institute of Neurology, London, UK
[8]UK Dementia Research Institute, London, UK
[9]Clinical Neurochemistry Laboratory, Sahlgrenska University Hospital, Mölndal, Sweden
[10]Department of Psychiatry and Neurochemistry, Institute of Neuroscience and Physiology, the Sahlgrenska Academy at the University of Gothenburg, Mölndal, Sweden
[11]Rugby Football Union (RFU), London, UK
[12]Arthritis Research UK Centre for Sport, exercise and osteoarthritis, University of Oxford, Oxford, UK
[13]Department of Neuroscience and Trauma, Blizard Institute, Queen Mary, University of London, London, UK

**Contributors** VG: co-principal Investigator. Drafted the protocol, collected comments and is corresponding author. DM: co-principal Investigator. Contributed to designing the protocol, reviewed intellectual content of this paper. HM: co-investigator. Advised on cognitive testing, biomarkers and clinical definitions, reviewed this paper for intellectual content. SC: co-investigator. Advised on cognitive testing and clinical definitions, reviewed this paper for intellectual content. SK: co-investigator. Advised on recruitment strategy, patient and public involvement (PPI) and reviewed this paper for intellectual content. CH: research assistant of the BRAIN Study. Recruits and collects informations on the study participants. DD: project administrator of the BRAIN Study. Coordinates the logistics of the study. NKA: co-investigator. Advised on recruitment strategy, patient and public involvement (PPI) and reviewed this paper for intellectual content. AM: co-investigator. Advised on biomarkers. MC: co-investigator. Advised on recruitment strategy, patient and public involvement (PPI) and reviewed this paper for intellectual content. MAMD: co-investigator. Advised on recruitment strategy, patient and public involvement (PPI), and reviewed this paper for intellectual content. HZ: co-investigator. Advised on biomarkers and reviewed this paper for intellectual content. NCF: co-investigator. Reviewed this paper for intellectual content. NP: co-principal Investigator. Contributed to designing the protocol, reviewed intellectual content of this paper.

**Funding** This study is funded by a grant from the Drake Foundation (www.drakefoundation.org/) to the London School of Hygiene and Tropical Medicine, with subcontracts to Queen Mary University of London (QMUL) and the Institute of Occupational Medicine (IOM).

**Competing interests** None declared.

**Ethics approval** Ethical approval was granted from the London School of Hygiene and Tropical Medicine (LSHTM) Ethics Committee (reference: 11634-2).

**Provenance and peer review** Not commissioned; externally peer reviewed.

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
