## [Reviewer comments · BMJ Open]

ARTICLE DETAILS

TITLE (PROVISIONAL)	BRain health and healthy AgeINg in retired Rugby union players, the BRAIN study – study protocol for an observational study in the UK
AUTHORS	Gallo, Valentina; McElvenny, Damien; Hobbs, Catherine; Davoren, Donna; Morris, Huw; Crutch, Sebastian; Zetterberg, Henrik; Fox, Nick; Kemp, Simon; Cross, Matthew; Arden, Nigel; Davies, Madeleine; Malaspina, Andrea; Pearce, Neil

VERSION 1 – REVIEW

REVIEWER	James Craig Brown University of Stellenbosch, South Africa
REVIEW RETURNED	18-Aug-2017

GENERAL COMMENTS	The proposed study, as currently described, will add enormously to this controversial and topical area and I look forward to following the results. As a result, I agree with authors' contention in the limitations section that there is insufficient time to wait for a prospective cohort study in this area. This justifies the cross-sectional study design. However my remaining main concern is around the absence of a control or non-rugby/non-collision sport group. My concern is that depending on how the recruitment takes place, there could be selection bias (opposite to what the authors predict in their "Limitations" section) in that those with cognitive/physical issues are more likely to join the study for a free assessment of these concerns. If this happened, this study might over-estimate the prevalence of these conditions. Whichever way participation might be affected – the way the authors predict or how I have described - would a control group not assist as they would be equally biased? Another strength of this study is the comprehensive range of tests to be performed, however could this induce fatigue in this age group and, if so, could this affect test performance? In general, editing/formatting could be improved. For example, passive voice is sometimes used; inconsistent use of capitals/lower case (Rugby Union vs rugby union); some in-text references are missing or incomplete; and at times the language could be clearer. In specific comments below, please note that the page number refers to that noted in the top right hand corner of the page (e.g. page 12 of 33). Specific comments Page 5, lines 3-4: 'no' instead of 'not'
--

	Page 6, lines 15-35: most of these symptoms seem plausible, except for 'weakness' and 'numbness'. Could this not include injuries such as 'stingers' to the shoulder? Should this not be more specific, such as 'generalised perception of weakness' or similar? Page 8, lines 26-30: this sentence is a bit unclear Page 8 (Genetics): would suggest including the systematic review of Panenka et al. 2017. Readers might benefit from the explanation of the three proposed methods of genetic modulation in this systematic review: (i) genetic modulation of inherent neurophysiology, (ii) polymorphisms associated with increased susceptibility to learning disability, (iii) polymorphisms associated with personality traits. Based on this final modulation (through personality traits), would it not be worth assessing this? See Mc Fie et al. 2017, JSAMS. Page 11, 45-48: But if the subjects have just signed informed consent or are aware of the study outcomes (as they should be), then they might VOLUNTEER information about previous concussions to the research team, thereby removing this blindness. Will this be noted, if it occurs, and analysed separately if there are sufficient of these cases? Page 14, lines 49/50: I'm unclear as to why a photograph of the hand is necessary? This seems out of place with the preceding and subsequent paragraphs, unless I have missed something. Page 14, lines 53/4: examines, not examined Page 14, lines 18/19: if a participant completes all the optional modules, how long will the assessment take? Page 14, Cognitive ability assessment: Could this test battery itself (and the order they are performed in) induce some sort of mental fatigue, thereby affecting results? Page 15, Neurological clinical assessment: would it be worth interviewing the spouse/partner of the study participants as they might have some insight into of neurological changes in their spouse/partner?
--	---

REVIEWER	Professor Patria Hume Sport Performance Research Institute New Zealand (SPRINZ), Auckland University of Technology (AUT), New Zealand
-----------------	--

REVIEW RETURNED	03-Sep-2017
-------------

GENERAL COMMENTS	Page 7 lines 32-34, correct typographical errors (likely a result of the pdf processing). Page 12 endnote referencing errors to be fixed on lines 5-6 and 18.
--

REVIEWER	Tom McMillan Institute of Health and Wellbeing University of Glasgow UK No Competing Interest
REVIEW RETURNED	19-Sep-2017

GENERAL COMMENTS	This is a timely study, there is significant interest in long term outcomes of head injury in sports given the high prevalence of repeated head knocks and uncertainty over whether these have long term consequences. The study is well informed and considers a range of outcomes in retired rugby players over 50 years, who as such are at a time when any long term sequelae are likely to become apparent. Some of the cognitive tests are outdated, in general this is not likely to be crucial unless using cut-offs, with the exception of the NART which will underestimate IQ unless scores are adjusted. Given the large number of associations that are to be investigated, allowance for type 1 error in analyses needs to be considered. The absence of a non-rugby control group may be perceived to be a limitation in design, although ideal controls for this population are difficult to achieve. It isn't clear if players reporting a history of concussion will be compared to those without (given the estimation that 50% will not report a history of concussion). As in all studies of this kind a key issue will be recruitment to the study to an extent that confers power to detect effects and in the ability of the study to consider any biases in recruitment which may lead to the finding of effects or their absence but which are not representative of the population of rugby players. The sample size calculation is based on cognitive tests only and it is not clear if the study is powered to detect effects on other measures. The fact that the sample has recently taken part in a study linked to health outcome in rugby players may lean towards optimism with regard to the authors reasonable recruitment predictions. In conclusion this looks to be a useful study which will add to the limited evidence base we currently have on long term effects of repeat head injury. I agree with the authors that the protocol will be useful to others who are developing projects on long term outcomes after sports injury.
---

VERSION 1 – AUTHOR RESPONSE

Reviewer: 1

Reviewer Name: James Craig Brown

General comments

The proposed study, as currently described, will add enormously to this controversial and topical area and I look forward to following the results. As a result, I agree with authors' contention in the limitations section that there is insufficient time to wait for a prospective cohort study in this area. This justifies the cross-sectional study design. However my remaining main concern is around the absence of a control or non-rugby/non-collision sport group. My concern is that depending on how the recruitment takes place, there could be selection bias (opposite to what the authors predict in their "Limitations" section) in that those with cognitive/physical issues are more likely to join the study for a free assessment of these concerns. If this happened, this study might over-estimate the prevalence of these conditions. Whichever way participation might be affected – the way the authors predict or how I have described - would a control group not assist as they would be equally biased?

Response: We agree that there will be selection effects operating in this study, and we are aware that we will not be able to estimate the prevalence of either cognitive decline or concussion among former Rugby players, within this study. Prevalence would be better estimated with a prospective cohort where player are recruited at the beginning of their career and followed up over time. With regards to the external comparison group, we would expect that this sample would be greatly affected by the 'healthy cohort effect' (not are only these all former elite rugby players, but a large proportion of them are Oxbridge graduates). We therefore decided not to have an external comparison group. Nonetheless, we replicated a simplified version of the 1946 birth cohort cognitive assessment which could be potentially used as reference, if needed. (The 1946 birth cohort was a representative sample of British children recruited at birth and followed continuously since). The risk of over- and/or under-representation of people with cognitive decline, on the other hand is real (as discussed, and further amended). However, this will not necessarily be associated with the number of concussions, and therefore it should not affect any internal comparison. Having a general population control group, unfortunately, would not modify the potential for selection bias.

Comment: Another strength of this study is the comprehensive range of tests to be performed, however could this induce fatigue in this age group and, if so, could this affect test performance?

Response: See answer below.

Comment: In general, editing/formatting could be improved. For example, passive voice is sometimes used; inconsistent use of capitals/lower case (Rugby Union vs rugby union); some in-text references are missing or incomplete; and at times the language could be clearer.

Response: Thanks, we have edited the text.

In specific comments below, please note that the page number refers to that noted in the top right hand corner of the page (e.g. page 12 of 33).

Specific comments

Page 5, lines 3-4: 'no' instead of 'not'

Page 6, lines 15-35: most of these symptoms seem plausible, except for 'weakness' and 'numbness'. Could this not include injuries such as 'stingers' to the shoulder? Should this not be more specific, such as 'generalised perception of weakness' or similar?

Response: The definition we used is adapted from the one proposed by the NIH (see https://commondataelements.ninds.nih.gov/Doc/TBI/F0482_Definition_of_Traumatic_Brain_Injury.doc), where a number of possible neurological symptoms are listed. We listed those in our definition, sometimes simplifying the lexicon for the lay public (i.e. weakness for paresis/plegia). This in our view reaches the best compromise between using the proposed definition, and make us understood by the participants.

Comment: Page 8, lines 26-30: this sentence is a bit unclear Page 8 (Genetics): would suggest including the systematic review of Panenka et al. 2017. Readers might benefit from the explanation of the three proposed methods of genetic modulation in this systematic review: (i) genetic modulation of inherent neurophysiology, (ii) polymorphisms associated with increased susceptibility to learning disability, (iii) polymorphisms associated with personality traits. Based on this final modulation (through personality traits), would it not be worth assessing this? See Mc Fie et al. 2017, JSAMS.

Response: Thanks for pointing out these important recent papers. We have added a more comprehensive view on the potential role of genetics in the modulation of the possible association between concussion and cognitive impairment

Comment: Page 11, 45-48: But if the subjects have just signed informed consent or are aware of the study outcomes (as they should be), then they might VOLUNTEER information about previous concussions to the research team, thereby removing this blindness. Will this be noted, if it occurs, and analysed separately if there are sufficient of these cases?

Response: This is a very good point. To date, the Research assistant was able to postpone any description of their concussion history to the very end of the interview. But, if and when this would prove not possible, the participant will be flagged, so to produce a sensitivity analysis. This is also now reflected in the text.

Comment: Page 14, lines 49/50: I'm unclear as to why a photograph of the hand is necessary? This seems out of place with the preceding and subsequent paragraphs, unless I have missed something.

Response: This is collected as part of the collaboration with our colleagues working on the musculoskeletal side in order to detect finger nodes, inflammation, and other signs of inflammatory osteoarthritis within this former elite sporting group

Comment: Page 14, lines 53/4: examines, not examined Page 14, lines 18/19: if a participant completes all the optional modules, how long will the assessment take?

Response: The Additional Optional Module should take no longer than 1 hour and half to be completed. We added also the cumulative time of 3 hours and 15 minutes

Comment: Page 14, Cognitive ability assessment: Could this test battery itself (and the order they are performed in) induce some sort of mental fatigue, thereby affecting results?

Response: The participants are responding well to the tests, and the allocated time for the session had not been exceeded in most cases. However, some participants have needed more time, but have never reported this to be a problem. The testing session is very varied with some questionnaires completed on a tablet, and some cognitive tests alternating with to physical assessments. To date, we have never interrupted a testing session for fatigue. Should this occur or become evident as a limitation within our study, we will report this alongside the study results.

Comment: Page 15, Neurological clinical assessment: would it be worth interviewing the spouse/partner of the study participants as they might have some insight into of neurological changes in their spouse/partner?

Response: We believe an objective neurological examination is more informative than some subjective information reported by the spouse. Moreover, not all participants come with their spouses (if they have one).

Reviewer: 2

Reviewer Name: Professor Patria Hume

Comment: Page 7 lines 32-34, correct typographical errors (likely a result of the pdf processing).

Response: Yes, these have been corrected, thanks

Comment: Page 12 endnote referencing errors to be fixed on lines 5-6 and 18.

Response: Thanks you. This has now been revised

Reviewer: 3

Reviewer Name: Tom McMillan

Comment: This is a timely study, there is significant interest in long term out come of head injury in sports given the high prevalence of repeated head knocks and uncertainty over whether these have long term consequences.

The study is well informed and considers a range of outcomes in retired rugby players over 50 years, who as such are at a time when any long term sequelae are likely to become apparent. Some of the cognitive tests are outdated, in general this is not likely to be crucial unless using cut-offs, with the exception of the NART which will underestimate IQ unless scores are adjusted. Given the large number of associations that are to be investigated, allowance for type 1 error in analyses needs to be considered.

Response: Thanks for this very relevant comment. We are aware of the potential risk of multiple comparisons, and we are currently developing a statistical analysis plan, which defines exactly the primary and secondary outcome measures to be considered. The study results will be interpreted according to the primary outcome measures.

Comment: The absence of a non-rugby control group may be perceived to be a limitation in design, although ideal controls for this population are difficult to achieve.

Response: As noted in response to reviewer 1, above about the external comparison group, we would expect that this sample would be greatly affected by the 'healthy cohort effect' (not only are these all elite rugby players, but many are Oxbridge graduates), therefore we decided not to have an external comparison group. The internal comparison of high vs. low exposed to concussion is much more

informative in this context. Nonetheless, we replicated a simplified version of the 1946 birth cohort cognitive assessment, which may be used as reference, if needed.

Comment: It isn't clear if players reporting a history of concussion will be compared to those without (given the estimation that 50% will not report a history of concussion).

Response: The plan is to compare highly concussed individuals with those with low or no prior concussions. However, the definition of concussion that we are currently using is slightly different from the definition that was used in the previous study. Therefore, it might be possible that a higher or lower proportion of participants will self-report concussion this time. Depending on the distribution of exposure, we will create an exposure matrix (i.e. unexposed, low exposed, and highly exposed) for exposure assessment.

Comment: As in all studies of this kind a key issue will be recruitment to the study to an extent that confers power to detect effects and in the ability of the study to consider any biases in recruitment which may lead to the finding of effects or their absence but which are not representative of the population of rugby players. The sample size calculation is based on cognitive tests only and it is not clear if the study is powered to detect effects on other measures.

Response: The power calculation is based on a continuous outcome measure. As outlined earlier, we are currently developing a statistical analysis plan, which defines the primary and secondary outcome measures to be used. The current sample will not be powered to detect association for a dichotomous outcome (e.g. a disease), and this analysis won't be attempted.

Comment: The fact that the sample has recently taken part in a study linked to health outcome in rugby players may lean towards optimism with regard to the authors reasonable recruitment predictions.

Response: All those approached in this study agreed to be re-contacted as part of the earlier study. The recruitment process to date is revealing very close to our estimates. If we do not succeed in recruiting the desired sample from the source population, we are planning to invite more potential participants through the RFU/Players' Union. In order to maximize our chances to recruit the expected numbers from the current population, we are in the process of producing a video interview to be launched in November, before the last recruitment round, to increase awareness of the study and ultimately increase participation rates.

Comment: In conclusion this looks to be a useful study which will add to the limited evidence base we currently have on long term effects of repeat head injury. I agree with the authors that the protocol will be useful to others who are developing projects on long term outcomes after sports injury.

Response: Thanks for this comment

VERSION 2 – REVIEW

REVIEWER	James Brown Stellenbosch University, South Africa
REVIEW RETURNED	12-Oct-2017
GENERAL COMMENTS	Thank you for responding so clearly to my comments. I am satisfied

	with the revised version and look forward to seeing the results of this fantastic study!
--	--